# Shear Wave Elastography and Shear Wave Dispersion Imaging in the Assessment of Liver Disease in Alpha1-Antitrypsin Deficiency

**DOI:** 10.3390/diagnostics11040629

**Published:** 2021-03-31

**Authors:** Marten Schulz, Moritz Kleinjans, Pavel Strnad, Münevver Demir, Theresa M. Holtmann, Frank Tacke, Alexander Wree

**Affiliations:** 1Department of Hepatology and Gastroenterology, Campus Virchow-Klinikum (CVK) und Campus Charité Mitte (CCM), Charité—Universitätsmedizin Berlin, 13353 Berlin, Germany; muenevver.demir@charite.de (M.D.); theresa.holtmann@charite.de (T.M.H.); frank.tacke@charite.de (F.T.); alexander.wree@charite.de (A.W.); 2Medical Clinic III, Gastroenterology, Metabolic Diseases, and Intensive Care, University Hospital, RWTH Aachen, 52074 Aachen, Germany; mkleinjans@ukaachen.de (M.K.); pstrnad@ukaachen.de (P.S.); 3Coordinating Center for Alpha1-Antitrypsin Deficiency-Related Liver Disease of the European Reference Network (ERN) “Rare Liver” and the European Association for the Study of the Liver (EASL) Registry Group “Alpha1-Liver”, 52074 Aachen, Germany

**Keywords:** alpha1-antitrypsin deficiency (AATD), shear wave elastography (SWE), shear wave dispersion (SWD)

## Abstract

Liver affection of Alpha1-antitrypsin deficiency (AATD) can lead to cirrhosis and hepatocellular carcinoma (HCC). A noninvasive severity assessment of liver disease in AATD is urgently needed since laboratory parameters may not accurately reflect the extent of liver involvement. Preliminary data exist on two-dimensional shear wave elastography (2D-SWE) being a suitable method for liver fibrosis measurement in AATD. AATD patients without HCC were examined using 2D-SWE, shear wave dispersion imaging (SWD) and transient elastography (TE). Furthermore, liver steatosis was assessed using the controlled attenuation parameter (CAP) and compared to the new method of attenuation imaging (ATI). 29 AATD patients were enrolled, of which 18 had the PiZZ genotype, eight had PiMZ, two had PiSZ and one had a PiZP-Lowell genotype. 2D-SWE (median 1.42 m/S, range 1.14–1.83 m/S) and TE (median 4.8 kPa, range 2.8–24.6 kPa) values displayed a significant correlation (R = 0.475, *p* < 0.05). 2D-SWE, ATI (median 0.56 dB/cm/MHz, range 0.43–0.96 dB/cm/MHz) and CAP (median 249.5 dB/m, range 156–347 dB/m) values were higher in PiZZ when compared to other AATD genotypes. This study provides evidence that 2D-SWE is a suitable method for the assessment of liver disease in AATD. The newer methods of SWD and ATI require further evaluation in the context of AATD.

## 1. Introduction

Alpha1-antitrypsin deficiency (AATD) is an autosomal codominant hereditary disease that mainly affects the lungs, causing emphysema or chronic bronchitis, while liver disease constitutes the second leading specific cause of AATD-related mortality. If liver affection is present, AATD can cause chronic hepatitis and liver fibrosis, leading to cirrhosis and hepatocellular carcinoma (HCC) [1]. A mutation in the alpha1-antitrypsin (AAT) gene (SERPINA1) leads to an incorrect tertiary structure and causes an accumulation of insoluble AAT in the endoplasmatic reticulum of hepatocytes [2]. This promotes hepatic inflammation and fibrogenesis that can yield to cirrhosis and HCC. AATD patients with a homozygous genotype (PiZZ) display a ~20 times increased risk for the development of liver cirrhosis, but hepatic involvement is also seen in the compound heterozygous phenotype (PiSZ) or to an even smaller degree in several other phenotypes [3] such as the heterozygous PiZ carriage, which is termed PiMZ. Currently, there is no approved specific therapy besides liver transplantation for AATD-associated liver disease [2]. Unlike in other chronic liver diseases such as viral hepatitis, AATD patients with liver affection do not necessarily display abnormal liver blood parameters [4]. This makes an accurate evaluation of the clinical state and course of AATD-related liver disease difficult. 

In order to assess liver disease severity and prognosis, knowledge of the presence and staging of liver fibrosis is crucial. Liver fibrosis detection was traditionally based on liver biopsy. Since this method is expensive and poses risks such as hemorrhage, pneumothorax or gall bladder injury, liver fibrosis assessment has widely shifted towards noninvasive liver stiffness measurement (LSM), which is a well-established method for detecting and quantitatively assessing liver fibrosis in chronic liver diseases such as viral hepatitis or nonalcoholic fatty liver disease (NAFLD) [5].

The most frequently used noninvasive LSM technique is transient elastography (TE). A 3.5 MHz probe sends a low-energy ultrasound (US) wave into the liver, and wave propagation is then evaluated by a receiver in the probe. The measured tissue stiffness is expressed in kilopascals (kPa). Point shear wave elastography (pSWE) is another LSM technique that is integrated into a conventional US machine and allows the selection of a region of interest (ROI) where the tissue elasticity is determined. Recent data suggest that in obese patients the skin-to-liver distance should be taken into consideration when interpreting pSWE results [6]. Furthermore, it has been shown in patients with chronic hepatitis C that LSM can be affected by elevated transaminases [7]. The pSWE measurement of the spleen has also shown promising results in predicting liver stiffness and portal hypertension [8,9,10,11]. Two-dimensional shear wave elastography (2D-SWE) is a more recent LSM method that allows the assessment of liver fibrosis by the real-time imaging of the propagation of shear waves in a focused region. A region of interest (ROI) can be placed in a colored elastogram in order to measure the liver stiffness, which is expressed in meters per second (m/S) or converted into kPa.

The determination of the so-called controlled attenuation parameter (CAP) via TE allows a noninvasive quantification of hepatic fat accumulation with high accuracy in fibrotic livers as well [12]. Attenuation imaging (ATI) is a newer technique that also uses US attenuation as a surrogate for the assessment of liver steatosis integrated in a conventional US machine. ATI allows for the visualization in B-mode US of the region in which the steatosis quantification is performed. This technique has shown promising results in NAFLD patients [13,14].

Even more recently, shear wave dispersion imaging (SWD) has been described as a new technique that allows assessing the dispersion slope of shear waves, which is related to tissue viscosity. Liver tissue viscosity is thought to be a surrogate for inflammation. Preliminary clinical studies suggest that necroinflammation of the liver can be measured via SWD [15,16]. However, published data on SWD is still scarce.

Several studies focused on the assessment of liver fibrosis in AATD patients. Liver biopsies in a cohort of 94 North American adults with the PiZZ genotype revealed a prevalence of significant liver fibrosis of 35.1% [17]. In a large study of European PiZZ adults who were examined with TE and noninvasive liver fibrosis tests, significant fibrosis (TE ≥ 7.1 kPa) was suspected in 20–36%. In this study, severe steatosis (CAP ≥ 280 dB/m) was present in 39% of PiZZ carriers, which was more frequent than in healthy controls [18]. Comparing the pSWE results of 41 AATD individuals with healthy participants, Diaz et al. found no difference in liver elasticity [19]. Reiter et al. compared the magnetic resonance (MRE), pSWE and 2D-SWE in 15 AATD patients. They found a high intermethod correlation and consistent identification of an increased shear wave speed in this cohort, providing preliminary evidence that elastography can be a helpful diagnostic tool in AATD liver disease [20].

The purpose of this study was to provide evidence on the applicability of 2D-SWE in the noninvasive evaluation of liver disease in AATD. Furthermore, our goal was to investigate the new techniques of SWD and ATI for the difficult assessment of liver affection severity in AATD. 

## 2. Methods

Enrolled patients had a known diagnosis of AATD confirmed by serum AAT levels and genotyping. Patients with viral hepatitis, HCC, cholestasis, excessive alcohol consumption or signs of right heart insufficiency were not included in the study. After fasting >4 h, participants received an abdominal ultrasound (US) including a grayscale US, Doppler measurement of the portal vein, spleen length measurement, 2D-SWE, SWD and ATI examination using the Canon Aplio i800 US system (Canon Medical systems Corporation, Otawara, Tochigi, Japan). US examinations were conducted by the same experienced operator with experience in liver ultrasound and sonoelastography (>6000 US, >2000 SWEs). Blood was drawn from all patients before US, and a routine liver panel was analyzed. The panel included, among others: Alanine aminotransaminase (ALT), aspartate aminotransaminase (AST), alkaline phosphatase (ALP), bilirubin, gamma-glutamyl transpeptidase (GGT), blood count, cholesterol, C-reactive protein (CRP), glucose, international normalized ratio (INR), serum ferritin and triglycerides. All patients gave written informed consent. 

### 2.1. 2D-SWE and SWD

The 2D-SWE value was defined as the median value of at least five measurements of good quality. A measurement was considered reliable when parallel lines in real-time propagation mode and a homogenous color filling of the 2D-SWE elastogram were present. Furthermore, an interquartile range/median (IQR/M) ratio ≤ 30% was defined as a quality criterion, as recommended [21]. Patients were examined in a supine position [22]. A ROI was then placed at least 1 cm below the liver capsule and 3–5 cm from the transducer in a right intercostal location, as depicted in Figure 1. Each measurement was conducted during a transient resting respiratory position while avoiding large vessels, according to the guidelines of the European Federation of Societies for Ultrasound in Medicine and Biology (EFSUMB) [22]. The reference cut-off values for 2D-SWE using the Canon US system from a histology-correlated study on 123 patients with diffuse liver disease were ≥7.8 kPa for fibrosis stage ≥ F2 (significant fibrosis), ≥9.4 kPa for fibrosis stage ≥ F3 (advanced fibrosis) and ≥12.2 kPa for F4 fibrosis (cirrhosis) [23].

Figure 1 shows the two-dimensional shear wave elastography (2D SWE) in a 64-year-old male with AATD.

The SWD examination included at least five measurements in the same manner as described above for the SWE measurements. Due to the scarcity of reference parameters, grading inflammation according to cut-off levels is difficult. Referring to the cut-off points proposed by Sugimoto et al., who recently examined a biopsy-controlled cohort of 111 individuals with suspected NAFLD with SWD, patients in our cohort could be classified as follows: inflammation grade A1: zero patients, inflammation grade A2 (≥9.9 (m/s)/kHz): five patients, inflammation grade A3 (≥A3 12.5 (m/s)/kHz): 24 patients [15].

### 2.2. ATI

Similar to the SWE and SWD measurements, the transducer was placed in a right intercostal position in a transient breath hold while the patient was in a supine position. Avoiding large vessels, a ROI was placed just below the orange-colored area, which represents the capsular artifact as depicted in Figure 2. If the coefficient of determination was ≥0.90, a measurement was considered valid; a median value of five acquisitions was determined. The ATI cut-off values referred to the biopsy-controlled parameters in a cohort with diffuse liver disease (S1 > 0.635, S2 > 0.700 and S3 > 0.745) [24].

Figure 2 shows the attenuation imaging (ATI) in a 58-year-old female with AATD.

### 2.3. TE/CAP

TE was undertaken using FibroScan (Echosens, Paris, France). This system has a choice of two probes (M-probe, 3.5 MHz and XL-probe, 2.5 MHz), and the XL-probe was selected in obese patients. US is used to follow up and measure the speed of the shear-waves generated in the liver tissue by the mechanical thump generated by a transducer mounted on the axis of a vibrator [21]. The results are expressed in kPa. Ten reliable measurements were performed in each patient (IQR/M ratio ≤ 30%, as recommended [22]). CAP was only calculated when the LSM measurement was valid, as described above, and the final CAP result was a median of 10 valid measurements. Since different disease-specific TE cut-off values have been described according to the underlying liver disease, we used previously recommended, etiology-unspecific cut-offs for LSM (F2 7.1–9.9 kPa, F3 ≥ 10 kPa and F4 ≥ 13 kPa) and CAP (>S0 ≥ 248 dB/m, >S1 ≥ 268 dB/m and >S2 ≥ 280 dB/m) [25,26]. 

### 2.4. Statistical Analysis

Data collection and statistical analysis were performed with IBM SPSS Statistics for Windows (Version 27.0. IBM Corp, Armonk, NY, USA), applying a Mann–Whitney U-test and Spearman’s rank correlation coefficient with a level of significance of *p* < 0.05 where appropriate.

## 3. Results

### 3.1. Descriptive Statistics of the Cohort

29 AATD patients were enrolled in the study, and the patient characteristics are displayed in Table 1. The genotypes were 18 PiZZ (62.1%), eight PiMZ (27.6%), two PiSZ (6.9%) and one was classified as PiZP-Lowell (3.4%). 15 patients were female (51.7%) and 14 were male (48.3%). The median age was 66.5 years (range 22–80 years). None of the included individuals had a known history of cirrhosis. Four patients (13.8%) received oxygen supply.

The SWE examination showed a median velocity of 1.42 m/S (range 1.14–1.83 m/S), accordingly 6 kPa (range 3.9–10.2 kPa). Using this LSM method, one patient was considered to have F2 fibrosis (8.3 kPa), and one patient had F3 fibrosis (10.2 kPa).

The SWD measurement displayed a median value of 14.2 m/S/kHz (range 10.3–18.5 m/S/kHz). The median ATI value was 0.56 dB/cm/MHz (range 0.43–0.96 dB/cm/MHz). Referring to the ATI cut-off values proposed by Bae et al., two patients had S1-steatosis and three patients had S3-steatosis [24].

Using TE, the median stiffness was found to be 4.8 kPa (range 2.8–24.6 kPa). Referring to the cut-off values of a study with etiology-unspecific liver disease, three patients were considered to have F2 fibrosis, one patient F3-fibrosis and one patient F4 fibrosis [18]. The median CAP value was 249.5 dB/m (range 156–347 dB/m). According to the cut-off values of a meta-analysis by Karlas et al., three patients had S1-steatosis, three patients had S2-steatosis and eight had S3-steatosis [26].

### 3.2. 2D-SWE Measurements in Patients with AATD Is Correlated with Established Predictors of Liver Fibrosis

The comparative analysis of 2D-SWE measurements revealed a significant correlation with the transient elastography (Spearman correlation 0.475 (m/S) and 0.456 (kPa) respectively, *p* < 0.05 for both). We also tested correlations in 2D-SWE measurements with established basic liver serum markers. The analysis revealed a significant correlation for 2D-SWE with the platelet count (Spearman correlation −0.591 (m/S) and −0.639 (kPa) respectively, *p* < 0.05 for both). The results of the Spearman’s correlation are displayed in Figure 3a–e. The SWD values correlated significantly with the spleen length (Spearman correlation 0.472, *p* < 0.05) and with the platelet count (Spearman correlation −0.508, *p* < 0.05). We expected a correlation of the SWD measurements quantifying liver tissue viscosity and being a surrogate for inflammation with liver transaminases. However, we found no correlation of SWD with ALT or AST (R = 0.015 and −0.058, respectively). 

Figure 3 shows the parameter correlation.

### 3.3. AATD Patients with PiZZ Genotype Show Increased Ultrasound-Based Liver Injury Parameters

As recently reported by Fromme et al., the PiZZ genotype harbors the highest predisposition for liver fibrosis/cirrhosis and liver cancer in AATD [27]. Hence, we speculated that a higher degree of liver injury could be detected noninvasively among PiZZ individuals in our cohort. Figure 4a–e shows higher statistically significant values of 2D-SWE, ATI, CAP and lower platelets in the PiZZ (*n* = 18) genotype than in patients with less severe AATD genotypes (*n* = 11). Comparing the median measurement values of patients with PiZZ to the results seen in the other genotypes, stiffness was higher in 2D-SWE (1.46 m/S versus 1.3 m/S and 6.45 kPa versus 5.1 kPa, respectively). Accordingly, platelets/nL were lower in PiZZ patients (156/nl, *n* = 11) versus non-PiZZ patients (240, *n* = 7). ATI and CAP (PiZZ *n* = 15, other *n* = 11) measurements displayed higher values of steatosis in the PiZZ group (0.57 dB/cm/MHz versus 0.52 dB/cm/MHz and 279 dB/m versus 216 dB/m). All levels of significance were *p* < 0.05.

Figure 4 shows the distribution of SWE, ATI, CAP and thrombocytes in AATD genotypes.

## 4. Discussion

Compared with other chronic liver diseases, AATD often displays a clinically unapparent course with asymptomatic patients who do not undergo liver biopsy [4]. Nevertheless, these “quiet” courses can lead to cirrhosis and HCC. Hence, there is a need for a noninvasive, accurate characterization of the extent of liver affection in AATD patients. Due to the relative scarcity of the disease, emerging measures for the noninvasive assessment of liver disease have not been systematically tested in AATD patients. In this study, we investigated the role of new methods for assessing fibrosis (2D-SWE), steatosis (ATI) and inflammation (SWD) in AATD patients.

2D-SWE measurements suggested a low percentage of significant fibrosis among our cohort. Referring to cut-off values from a histology-correlated study in 123 patients with various liver diseases [23], one patient (3.4%) was considered to have F2-fibrosis and one patient had F3-fibrosis. The TE examination revealed three patients with F2-fibrosis, one with F3- and one with F4-fibrosis. The two patients with elevated stiffness in 2D-SWE (F2 and F3) were the same patients who displayed higher TE levels. Accordingly, TE and 2D-SWE showed a statistically significant correlation, as displayed in Figure 3. The correlation of the well-established fibrosis-assessment method TE with SWE supports prior evidence from a smaller cohort [20], suggesting that 2D-SWE is a suitable method for fibrosis detection and staging in AATD.

As in other liver diseases, markers that allow an accurate assessment of the severity of liver disease besides the detection of fibrosis are needed in AATD. The noninvasive determination of necroinflammation plays a crucial part in this field, especially the distinction from fibrosis. SWD is a new technique that has shown promising results in anecdotal clinical application, and no data on its applicability in AATD have been published so far. Since this method is not established, the sparsity of published studies renders the comparison and interpretation of measured values difficult. The SWD results in our cohort displayed high dispersion slopes. Referring to the cut-off values proposed by Sugimoto et al. for a cohort of individuals with suspected NAFLD, no patient in our cohort displayed normal or SWD values of mild inflammation [15]. However, the correlation with ALT and AST levels, which are markers for liver inflammation in various chronic liver diseases, did not show a statistically significant association with SWD. We observed a correlation of SWD with spleen length and platelet count, which are known as markers for fibrosis rather than for inflammation. Notably, SWD might be affected by AAT accumulation, which constitutes the hallmark of AATD [2]. Further research involving the biopsy-correlation with SWD of different chronic liver diseases is needed for a better understanding of this promising technique. In addition, disease-specific cut-off values need to be implemented for all presented diagnostic tools. This is especially important for AATD since established cut-offs have been mainly investigated in cohorts without lung disease. 

Steatosis quantification using ATI detected steatosis in five patients, while the CAP measurement showed steatosis in 13 patients. Accordingly, no significant correlation was found when comparing ATI and CAP, which also calls for a further evaluation of both methods with histologic controls in a larger AATD population.

It is known that the PiZZ genotype predisposes for a more severe liver disease [3]. Of note, in our cohort, PiZZ patients displayed higher values in 2D-SWE than patients with other genotypes (1.46 m/S versus 1.31 m/S and 6.5 kPa versus 5.2 kPa, respectively). Furthermore, the ATI and CAP measurements revealed a higher hepatic fat accumulation in PiZZ genotype patients. These findings also indicate that 2D-SWE and ATI are feasible methods for discriminating more severe liver affection in AATD.

The limitations of this cross-sectional monocenter study include a relatively small cohort and the lack of a histological correlation due to clinically stable patients without a current indication for liver biopsy. The correlations in our study are significant but rather weak and should be interpreted as indicative in the context of a small cohort size. Furthermore, most enrolled patients did not have advanced fibrosis; however, the composition of the cohort reflects the fibrosis burden seen in AATD individuals. 

Taken together, this study contributes further evidence that 2D-SWE can be a helpful noninvasive tool in assessing liver disease in patients with AATD. The new techniques of ATI and SWD are promising and require further evaluation.

## Figures and Tables

**Figure 1 diagnostics-11-00629-f001:**
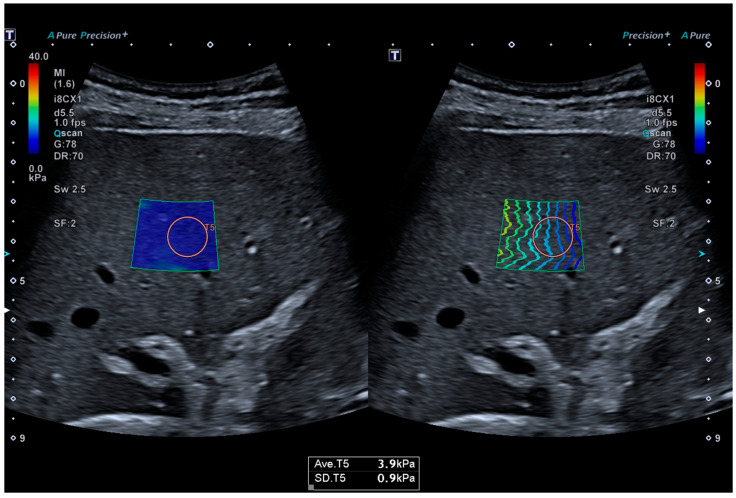
The propagation of the shear waves in the liver tissue is visualized as color-coded (**left side**) and in parallel lines (**right side**). The measured liver stiffness is 3.9 kPa, indicating the absence of significant fibrosis.

**Figure 2 diagnostics-11-00629-f002:**
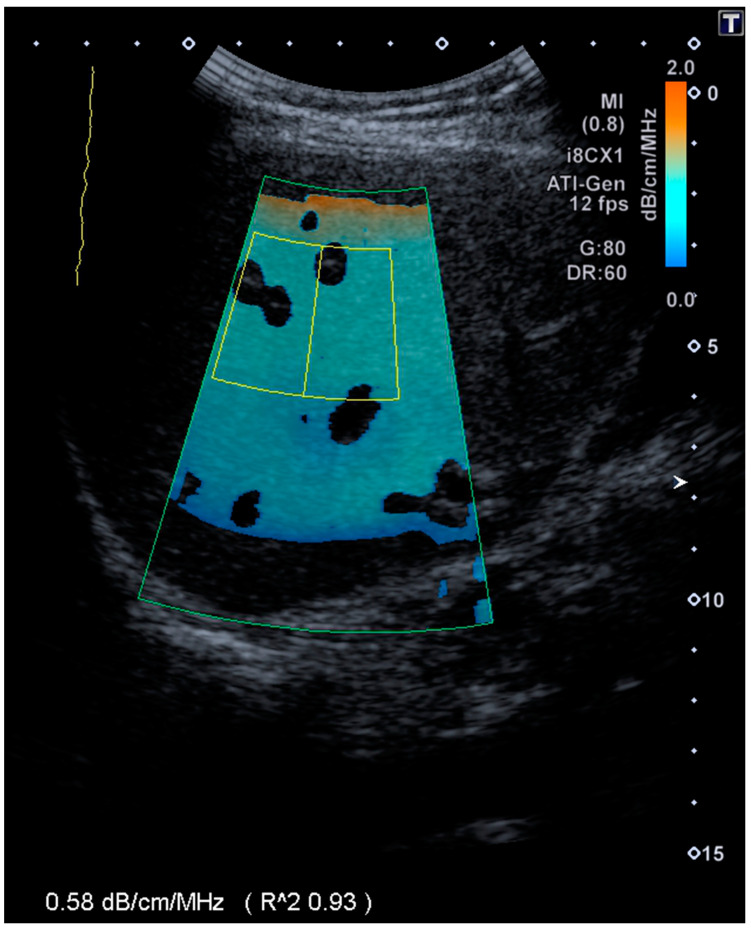
A region of interest (ROI, yellow marked area) is placed below the capsular artifact, which is visualized in orange color. The measured attenuation coefficient is 0.58 dB/cm/MHz, indicating the absence of steatosis.

**Figure 3 diagnostics-11-00629-f003:**
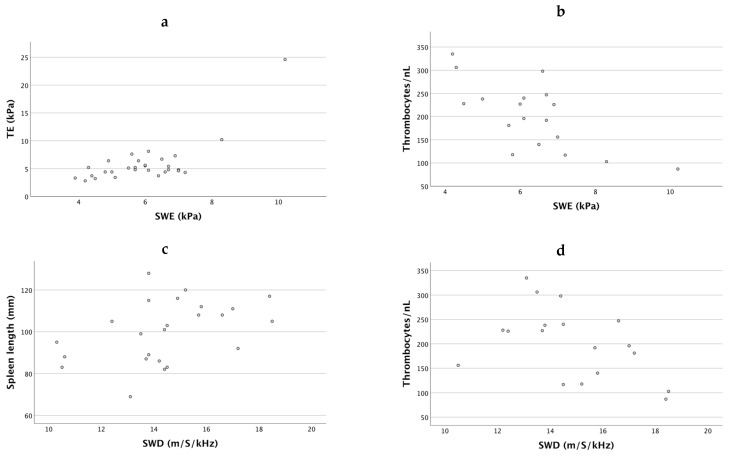
(**a**,**b**): Spearman’s correlation of TE (kPa) and thrombocytes/nl and 2D-SWE (kPa). (**c**,**d**): Spearman’s correlation of the spleen length (mm) and thrombocytes/nL and SWD (m/S/kHz) All *p*-values < 0.05. (**e**): no correlation between ALT/AST and SWD. Abbreviations: ALT: alanine aminotransaminase; AST: aspartate aminotransaminase; GGT: gamma-glutamyl transpeptidase; SWE: shear wave elastography; SWD: shear wave dispersion; TE: transient elastography.

**Figure 4 diagnostics-11-00629-f004:**
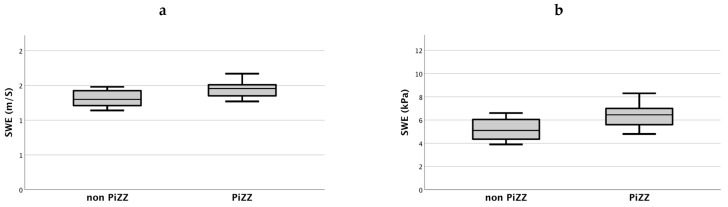
(**a**–**e**): Bars on the right-hand side display the PiZZ genotype (*n* = 18), and bars on the left-hand side show other AATD genotypes (*n* = 11). The measurement of SWE, ATI, CAP and thrombocytes/nl discriminate PiZZ from other AATD genotypes. All *p*-values < 0.05. SWE: shear wave elastography; ATI: attenuation imaging; CAP: controlled attenuation parameter.

**Table 1 diagnostics-11-00629-t001:** Patient characteristics, LSM findings and steatosis assessment.

Characteristics	Patients (*n* = 29)
Genotype	18 PiZZ
	8 PiMZ
	2 PiSZ
	1 PiZP-Lowell
Gender	15 Female (51.7%)
	14 Male (48.3%)
Age	Median 66.5 Years, Range 22–80 Years
Laboratory Findings:	
Total Bilirubin	0.49 mg/dL, Range 0.1–1.16 mg/dL
ALT	23 U/L, Range 6–87 U/L
AST	25 U/L, Range 15–94 U/L
GGT	24 U/L, Range 10–155 U/L
Platelet Count	211/nL, Range 87–335 U/L
US Findings:	
PV Maximum Velocity	20.9 cm/S
Spleen Length	102 mm
2D-SWE	1.42 m/S, Range 1.14–1.83 m/S
	6 kPa, Range 3.9–10.2 kPa
SWD	14.2 m/S/kHz, Range 10.3–18.5 m/S/kHz
TE	4.8 kPa, Range 2.8–24.6 kPa
ATI	0.56 dB/cm/MHz, Range 0.43–0.96 dB/cm/MHz
CAP	249.5 dB/m, Range 156–347 dB/m

Abbreviations: LSM: liver stiffness measurement; ALT: alanine aminotransaminase; AST: aspartate aminotransaminase; GGT: gamma-glutamyl transpeptidase; US: ultrasound; PV: portal vein; 2D-SWE: two-dimensional shear wave elastography; SWD: shear wave dispersion; TE: transient elastography; ATI: attenuation imaging; CAP: controlled attenuation parameter.

## Data Availability

The data are not publicly available due to privacy restrictions.

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
