# Peer review of "Shear Wave Elastography and Shear Wave Dispersion Imaging in the Assessment of Liver Disease in Alpha1-Antitrypsin Deficiency"

_diagnostics, 2021, doi:10.3390/diagnostics11040629_

Round 1

Reviewer 1 Report

This article is well-written and argued and its advisable for publication.

Reviewer 2 Report

Methods

  • US - elastography operator was the same?
  • Did the authors also employed IQR/M values to define reliability of measures?
  • Please define patients position during elastography evaluation
  • Please define if data distribution was studied and what test was employed. Also, p-value was two tailed?

Results

  • Figure 3: Are you sure those date are correlated? Data dispersion is relevant from what appear in the figures.
  • Figure 4: the term correaltion is misused. You are simply reporting difference in means/medians between two groups, you are not studing correlation.

Discussion:

  • Line 249-251: when referring to NAFLD I suggest citing this recent paper on pSWE (doi: 10.3390/diagnostics10100795) that reported how LS is affected by skin-to-liver distance in terms of fat tissue and how it may affect reliability.
  • Lines 252-253: the authors should compare their results on transaminases level with this recent paper on another MDPI journal doi: 10.3390/microorganisms8030348
  • Line 254-256: similar findings were found in these papers, please cite them in order to strengthen your results (doi 1: https://doi.org/10.1016/j.aohep.2019.09.004 - doi 2: https://doi.org/10.1016/j.aohep.2019.03.004 - doi 3: 10.1007/s40477-020-00456-9 - doi 4: https://doi.org/10.1016/j.aohep.2020.07.004)

Reviewer 3 Report

I was asked to review the paper entitled “Shear wave elastography and shear wave dispersion imaging in the assessment of liver disease in Alpha1-antitrypsin deficiency”. It is an interesting, well written paper.

However, there are some changes that should be made, as follows

Major comments:

  1. The number of patients is rather small and they were not evaluated by liver biopsy as a reference method, but I think that it cannot be helped due to the rarity of the disease, while the biopsy is less and less accepted by the patients, due to the development of non-invasive methods.
  2. Material and methods section: cut-offs used for SWD, ATI, LSM by TE as well as for CAP should be mentioned, the same as you did for 2D-SWE.
  3. Material and methods, TE/CAP section. You say: “With the probe in a right intercostal position, a US signal is emitted transcutaneously and its propagation in the tissue is measured by a receiver in the probe which is related to liver tissue stiffness”. It is incorrect. US is used to follow up and measure the speed of shear-waves generated into the liver tissue by the mechanical thump generated by the transducer mounted on the axis of a vibrator. Please correct. See as reference the WFUMB guidelines and recommendations for clinical use of ultrasound elastography: part 1: basic principles and terminology http://dx.doi.org/10.1016/j.ultrasmedbio.2015.03.009.
  4. Material and methods, TE/CAP section. You mentioned a SR ≥ 60% as a quality criterion. The latest guidelines do not recommend its use anymore. https://doi.org/10.1016/j.ultrasmedbio.2018.07.008 and DOI http://dx.doi.org/10.1055/s-0043-103952
  5. Was it a prospective or a retrospective study? You said that signed informed consent was obtained. For what? Elastographic measurements, data acquisition and analysis, participation in the study
  6. The correlations of 2D-SWE and TE and ATI and CAP, respectively, even if significant are rather weak, I think it should be mentioned and discussed.
  7. In the discussion section I think that the influence of right cardiac insufficiency as a confounding factor for stiffness measurements should be mentioned. Did you have such patients in your cohort?

Minor comments:

  1. Introduction, row 80 and 82: You should explain what significant fibrosis and severe steatosis, mean
  2. “2.1.2 D-SWE and SWD”, I think you meant 2.1.2 2D-SWE and SWD
  3. “3.2.2 D-SWE measurements in patients with…” I think you meant 3.2.2 2D-SWE measurements in patients with…

Round 2

Reviewer 2 Report

The authors have edited the manuscript according to the reviewers comments.

The manuscript can now be accepted for publication.